# Announcement Signals and Automatic Braking Using Virtual Balises in Railway Transport Systems

**DOI:** 10.3390/s22051943

**Published:** 2022-03-02

**Authors:** Enrique Santiso, José A. Jiménez, Manuel Mazo, Cristina Losada, Francisco J. Rodríguez

**Affiliations:** Department of Electronics, University of Alcalá, 28805 Alcalá de Henares, Madrid, Spain; enrique.santiso@uah.es (E.S.); jose.jimenez@uah.es (J.A.J.); manuel.mazo@uah.es (M.M.); cristina.losada@uah.es (C.L.)

**Keywords:** railway transport, railway transport safety, virtual balise, digital map, automatic train protection systems, ASAB-VB

## Abstract

In rail transport, various automatic protection systems are available to ensure the safe operation of trains and to facilitate automation and optimization tasks. For this purpose, a set of physical balises is used, which are placed at fixed points along the railway track. Based on the information provided by these balises, different information is displayed to the driver, and control actions are generated. The use of physical balises located at fixed points does not allow for automatic protection actions on sections of track where they are not installed. This is a major drawback as in many cases, temporary automatic protection actions are necessary on sections of the railway line without balises due to various circumstances (work on the track, accidents, etc.). To solve this problem, this paper presents a solution called announcement signals and automatic braking using virtual balises (ASAB-VB). This proposal allows the incorporation of virtual balises at points on the track where it is necessary to temporarily perform automatic protection actions. For this purpose, the ASAB-VB system allows obtaining the train position in real-time and storing a digital map of the track that will be made by each train. This digital map includes geographic information about the balises (both physical and virtual ones) located on the track. At the same time, the train position is obtained by merging the information provided by a GNSS, an odometer, and an inertial system (gyro and accelerometers).

## 1. Introduction

Railway transport is becoming increasingly important worldwide, mainly because it is one of the safest modes of transport when compared to other means of passenger transportation. For example, according to the Worldwide Railway Organisation (UIC), the accident rate is almost 200 times higher for road than for railway transport. This, together with the contribution of railways to sustainable transport policies, justifies the promotion of this mode of transport in most countries. All this is also favored by the incorporation of new technologies, which allow increasing the safety performance of this means of transport. Additionally, thanks to the incorporation of these technologies, train crashes are becoming rarer in the European Union. Recently published Eurostat data confirm this positive trend. In 2019, there were 516 significant railway accidents with 802 fatalities in the EU, down from 853 deaths in 2018 [1]. According to the “Report on Railway Safety and Interoperability in the EU-2020” [2], the fatality risk for a train passenger is a quarter of the risk for a bus/coach passenger. The use of individual transport means, such as passenger car, carries a substantially higher fatality risk: car occupants are almost 50 times more likely to die than train passengers traveling the same distance. The fatality risk for an average train passenger is now about 0.05 fatalities per billion passenger-kilometers, making it the safest mode of land transport in the EU.

In the field of railway transport, automatic train control (ATC) and automatic train protection (ATP) systems enable the safe operation of trains, as well as the automation and optimization of their functioning. Train protection systems perform a variety of actions, such as preventing the train from running a red light, controlling train speed, warning the train driver in advance, or activating emergency brakes in case of danger. In Europe, each country has developed its automatic protection system, the most significant difference being the type of connectivity they have between the track and rolling stock (induction, radio, etc.) as well as the speed control they perform (punctual or continuous). However, the establishment of trans-European rail networks has led to the development of the European Rail Traffic Management System (ERTMS), which ensures interoperability within the European rail network. ERTMS consists of a train control system (European Train Control System—ETCS) and a communications system (GSM Railways—GSM-R) to carry out a continuous transmission via-train, each time the train passes over a Eurobalise (balises installed on the track). Within the ERTMS technology, there are three levels of complexity in its operation, and all of them provide continuous on-board supervision. In Spain, different automatic train protection systems coexist, with the digital ASFA (Automatic Signal Announcement and Braking) being one of the most widely used due to its characteristics.

Most automatic train protection systems (ERTMS, ASFA, etc.) use physical balises (PB) installed at known geographical coordinates along railway tracks, together with a detector (receiver) system onboard the train. When the train passes the balises, the onboard system recognizes their status and displays information in the cabin on the appearance of the side signals and the maximum speed of the section to facilitate driving. Moreover, it can control train speed and driver actions, activating emergency braking in the event of detecting abnormal driving situations. As an example, in Spain, the digital ASFA system uses passive physical balises installed on the track, along the railway lines, and a detector system (receiver) onboard the trains. When the train passes over the balises, the receiver system recognizes their status by means of a magnetic coupling balise-receiver and, from this information, obtains the speed restrictions on the track sections following the balise location points. The onboard system includes in its receiving part an oscillator that oscillates at a certain frequency in the absence of balise detection (free oscillation frequency). When the train passes over the balises located on the track, the oscillator changes its free oscillation frequency to the resonance frequency forced by the balise circuit. The digital ASFA is designed to detect up to nine different frequencies, in the range of 55–115 kHz. Each frequency corresponds to what is known as the “balise aspect” (in the case of the digital ASFA, up to 9 “balise aspects” are considered: L1—stop announcement/pre-stop announcement/etc., L2—conditional free lane, L3—free lane/protected level crossing, etc., and for each one a braking curve profile is set. In addition, there is vertical signalling which also informs about the status of the balise.

Regarding the train positioning, several proposals use GNSS and inertial systems. Thus, in [3] a study was carried out on the reliability of low-cost GNSS receivers for estimating the position of trains. The authors of [4] propose a data fusion scheme using GNSS and odometry to locate trains and detect if the integrity of a railway convoy is accidentally broken (some wagon is uncoupled from the convoy). Whereas [5] describes a train location system on low traffic lines that is based on GNSS and in which the train communicates with a central system, via radio, to receive running orders based on its position. Systems for accurate train location based on the fusion of information provided by GNSS, odometers and inertial systems are presented in [6,7]. An integrated INS/GNSS train positioning system with odometric data is proposed in [8]. Finally, a detailed analysis of the most relevant works related to data fusion between GNSS and inertial system can be found in [9].

About the use of PBs as milestones to correct the train position each time one of them is read, there have also been works carried out using digital maps to improve the accuracy and reliability of the information provided by satellites. In this context, the authors in [10] point out the relevance of digital maps within the map-matching process required in train positioning systems based, in this case, on GALILEO. The authors of this work highlight the importance of the digital map in positioning systems in safety-critical railway applications. In [11], positioning methodology for train navigation applications based in a satellite system called BDS (BeiDou navigation satellite system), inertial system, odometer, and map-matching is proposed to solve the problem of positioning when satellite system outages occur when trains pass through signal obstructed areas, such as under bridges, inside tunnels, and through deep valleys.

The virtual balise (VB) concept arises in the context of ERTMS to obtain an improved ERTMS. The virtual balise concept consists of replacing a physical balise by a virtual analogue, thus reducing the number of physical balises on the track and resulting in cost savings [12]. The principle of operation is based on storing the list and coordinates of virtual balises and using them together with GNSS positioning information to know the location of the train [13]. The VB concept has been successfully investigated in many research projects, e.g., ESA3InSat [14], GSA ERSAT-EAV [15], GSA RHINOS [16], and GSA STARS [17]. The ASAB-VB system is intended for conventional lines with low traffic density where ERTMS is not available. Therefore, the concept of VB is different in this application, where VBs are used to facilitate the incorporation of temporary speed limits (TSL) on the railway track.

Essentially, all the previous works are based on how to solve the problem of knowing the exact position of the train as reliably as possible, using different sensory systems (GNSS, odometric, inertial systems, radar, etc.) and, in some cases, including digital maps. This paper presents a novel system called announcement signals and automatic braking using virtual balises (ASAB-VB) that can obtain the position of the train in real-time, as well as store the digital map including the coordinates of all the balises detected along the route. The train position and the digital map are used to generate virtual balises when it is necessary to temporarily perform automatic protection actions. The main contributions of the proposed system lie in (i) the fact that it allows the incorporation of virtual balises (VB) that complement and/or replace the PBs and (ii) the incorporation of virtual marks (VM) used to correct the error accumulated by the odometric system. It is based on the train position obtained by the fusion of GNSS information with odometric and inertial systems. Additionally, the system may also be used to obtain information on the status of the physical balises (operational balises, absence of balises, damaged ones) with the consequent saving in maintenance costs. There are no previous works proposing the use of VBs and VMs for the purposes indicated in this paper.

The rest of this paper is organized as follows: Section 2 gives an overview of the proposed solution, Section 3 shows the architecture of the ASAB-VB system, Section 4 details the positioning algorithm, Section 5 presents the main results obtained during the testing phase and, finally, Section 6 presents the conclusions.

## 2. General Overview of the Proposal

As explained above, most current automatic train protection systems use physical balises located at fixed points along the railway track. This means that it is not possible to carry out automatic protection actions on sections of the track where these PBs do not exist. This is a major drawback, as in many cases it is necessary to carry out temporary automatic protection actions, due to various circumstances (maintenance and repair work on the track, accidents, etc.) on sections of the railway track where there are no physical balises. This is one of the main reasons that have led the authors of this work to develop a new system that complements the current automatic protection systems and incorporates improvements from the point of view of both safety and flexibility. Among the advantages of this proposal, called Announcement Signals and Automatic Braking using Virtual Balises (ASAB-VB), the following are worth mentioning:Incorporation of the digital map with VBs based on the train position obtained by the fusion of GNSS information with odometric and inertial systems. These VBs either complement, replace, or both, the physical ones.Facilitating the incorporation of temporary speed limits (TSL) on the railway track by making use of these VBs. These TSLs should be established when unexpected situations occur on the track (automatic protection actions): maintenance, accidents, etc.Obtaining information on the status of the PBs (operational balises, absent balises, or damaged ones) with consequent savings in maintenance costsUse of VMs to correct the error accumulated by the positioning system.

To achieve all these functionalities, the ASAB-VB system requires an infrastructure on the ground and on-board railway units, to know the position of the train within the railway line at any time, as well as the location of the PBs on the track.

The ASAB-VB system (patent number ES 2418929 A1, [18]) has been designed to complement the digital ASFA, although it could be a complement to other automatic train protection systems, such as the ERTMS [19]. Under this premise, Figure 1 shows the general configuration of the proposed ASAB-VB system.

The ASAB-VB system includes the following elements in each railway track (see Figure 1):Physical balises (PBs): these are the ones used in the digital ASFA system. These balises are read by the train as it passes over them. Depending on the type of balise (so called “balise appearance”) they have a certain effect on the signalling and thus on the train speed limitation. It should be noted that the external appearance of the PBs is the same regardless of their type or “balise appearance”. The train’s balise detection system consists of a resonant circuit, that oscillates at a certain frequency (permanent frequency, PF). When the detector passes over a balise, it oscillates at the resonant frequency of the circuit set in the beacon, to which it is tuned by inductive coupling, receiving the corresponding “balise appearance”. Figure 2 shows the external appearance of a PB installed on a track.

Virtual balises (VBs): are balises that are established virtually at certain kilometer points where the passing of the train through them also has a certain effect on signalling and thus on the train speed. The need for virtual balises may be due to the requirements to establish speed limits at points on the track where no physical balises exist, or to replace damaged or stolen physical balises.Virtual marks (VMs): only serve as a complement to the train positioning system. Specifically, VMs are used to correct the error accumulated by the positioning system, as described below. So, they do not affect signalling.

The general flow chart of the Announcement Signals and Automatic Braking using Virtual Balises system (ASAB-VB) is shown in Figure 3, where two different stages are contemplated: before the start of the route (static train), and during the route.

As can be seen in Figure 3 before the train starts a certain route, the following two actions are performed:Obtaining the track on which the train is located: before starting the route, it is necessary to determine the track on which the train is located. For this purpose, a positioning system based on differential global navigation satellite system (DGNSS) is used.Loading the track balise map (TBM) in the on-board unit: before the start of a route, the track balise map (TBM) is stored in the on-board unit, which includes geographic information on the balises (PB and VB) and virtual marks (VM). The TBM includes the following information about each balise (PB and VB): the track on which they are located, the kilometer point of the balise position on the track (KPB and KPVB), and its appearance (effect on signalling). The information included in the TBM on the VMs is only the GNSS coordinates (latitude and longitude) and kilometer point of the virtual mark position on the track (KPVM). The TBM transfer from the geographic information system (GIS) to the train can be done via radio (3-4-5G/Wimax), or can be loaded locally via USB if communications are not available. If necessary due to any unforeseen situation (maintenance requirement, accident on the track, etc.), the transferred file (TBM) can be modified during the route if there is radio communication with a ground station (e.g., at an intermediate station on the route).

Once the route has started, the three most relevant activities are:Positioning: the position of the train on the track is defined by the track number on which it is located (TN) and the kilometric point of the train position on the track (KPT). To have full availability of the train position, these data are continuously updated by merging the data obtained by the odometric and inertial system. The error accumulated by this estimation is reduced each time a PB is detected, or when the GNSS estimated position matches that of a KPB.Generation of virtual balises (VBs): when the kilometer point corresponding to a virtual balise (KPVB) stored in the TBM is reached, the VB is generated.Detected track balises map (DTBM): during the route, a map of physical balises called a detected track balises map (DTBM) is generated, in which the PBs detected during the route are registered, their GNSS coordinates, kilometric point, and the time while their detection was maintained called time of balise (t_B_). This time is indicative of the quality of the detection of the balise. At the end of the route, the DTBM is transferred to the control center for analysis.

The information provided by the positioning system, together with the track balises map (TBM), makes it possible to know not only whether or not the physical track balises are detected correctly or whether they are at the correct kilometric points (KPT), but also whether there are undetected balises, their detection quality, etc. For this purpose, once the route is completed and the DTBM has been transferred to the ground system, the information associated with the physical balises contained in the DTBM is matched with the TBM, previously loaded onto the train.

## 3. Architecture of the ASAB-VB System

To achieve the objectives, the ASAB-VB system requires an infrastructure on the ground and onboard railway units. Figure 4 shows the most important modules of track-side and onboard subsystems.

The ground infrastructure consists of the following elements:Geographic information system (GIS): generates the map with the balises of each route (track balise map, TBM).Control, analysis, and registration unit: in charge of managing communication with the train, recording incidents detected along the route, and updating TBMs dynamically in those cases in which, once a route has started, it is necessary to include new VBs. It is also responsible for the comparison between the TBM and the detected track balises map (DTBM), which contains the balises detected during a route and their location. This comparison makes it possible to obtain the possible incidences of the route: absence of balises at the points defined in the TBM, time of detection of balises, balises location errors, and balises detected on the route and not contemplated in the TBM.GNSS: allows differential positioning of the train. In the case described in this paper, before starting a certain route, the track on which the train is located must be known, and for this, it is necessary to use a differential GNSS.Communications module: manages communications with the train. In this case, it is based on 3-4-5G/Wimax communication.

The infrastructure onboard mobile unit consists of the following elements:
Control unit: manages all the operations of the elements of the ASAB-VB system installed onboard the mobile units. Specifically, assuming an onboard digital ASFA system, its functions are:
−Manage communications between the elements of the mobile unit, as well as between the ASAB-VB and ASFA digital systems.−Generate the virtual balises (VB) towards the digital ASFA system in case of failure in the detection of physical balises (PB’s) if this functionality is enabled.−Process the TBM to create, from the train position (obtained by the “positioning system” module), the corresponding virtual balises at the corresponding kilometer points (KPTs).−Obtain information on the time instant at which a PB has been detected by the ASFA system.−Transmit to the ground infrastructure the detected track balises map (DTBM). As explained above, the DTBM contains the physical balises detected during the route and their location.Positioning system: determines the position of the train from the integration of the different sensorial systems: GNSS, the detector of balises, and a solution that integrates odometry and inertial system by means of a Kalman filter.Communications module: responsible for managing communications between the train and the ground infrastructure.

## 4. Positioning Algorithm

As already explained, the train positioning system is one of the most important aspects of the ASAB-VB system. This module makes it possible to generate the VBs at the desired points along the route and to confirm the status and correct location of the PBs. All this is in order to establish the corresponding actions on the signalling and, thus, on the speed of the train on the desired sections of the route.

The solutions for the initial positioning of the train before starting a route and during the route are different.

### 4.1. Initial Positioning

Before starting a route, the track on which the train is located has to be determined. For this purpose, a differential GNSS (DGNSS) is used, with an instantaneous correction obtained through wireless communication (Wi-Fi/Wimax) with the terrestrial infrastructure. This positioning system ensures that the train’s position is obtained with an accuracy of less than 1 m. Since the distance between the longitudinal axes of two tracks is between 3.8 and 4 m, an error of less than 3.8/2 = 1.9 m would be enough to guarantee the discrimination of the track on which the train is located. This error will be identified by ε_in_, therefore ε_in_ ≤ 1.9 m.

### 4.2. Positioning during the Route

During the route, the ASAB-VB system performs positioning by fusing the odometric and inertial system, corrected with GNSS and PBs. Figure 5 shows the general architecture of the positioning system during the route. As can be seen, two methods of position measurement are envisaged: (1) a GNSS and a system that integrates odometry and inertial information (which will be identified in what follows by “Kalman-position”); and (2) additionally, there is an input from the control unit that provides information (kilometer point of the balise position on the track (KPB) and time of balise (t_B_)) when a PB is detected.

Maximum accuracy in the train positioning is only necessary at points where there is some type of signalling, and at these points, the most important thing is relative positioning, since the signals affecting the movement of the train generally appear in pairs at pre-established distances to establish a given restriction. Within the distances between the two signals, the train must carry out certain actions, and if it fails to do so, automatic braking would be applied. For these reasons, the position shall be periodically estimated by “Kalman-position” and corrected using the error in the points before the location of the signals.

The train positioning can be reduced to a one-dimensional positioning, where it is sufficient to know the longitudinal coordinate of the track on which the train is located. To keep this coordinate updated, known as the kilometer point of the train position on the track (KPT), the fusion of the odometer and inertial system data, carried out by means of a Kalman filter, is used. The ASFA standard [20] requires positioning systems with these characteristics to have an error of less than 5% over distances of up to 800 m. Some works comfortably meet these specifications, with errors less than 5 m in 1 km [21], so it can be assumed that the odometric-inertial system will keep an estimation of the position with an error less than these values.

In the case of tunnels, in the absence of GPS coverage, the positioning system is based on the information provided by the odometric system. It should be noted that in most tunnels there are physical balises that allow correcting the odometric error, being able to position a VM at the closest point to the tunnel entrance at a point where GPS coverage is good. Correcting the odometric error when GPS coverage is expected to be lost in the vicinity of the VMs is precisely the purpose of VMs.

### 4.3. Reduction of the Positioning Error during the Route

ERTMS systems place balises at every kilometer to reset the above-mentioned errors. The system proposed in this paper is aimed at conventional lines with low traffic density, where this signalling does not exist. To reduce the odometric-inertial system error, the information contained in the TBM, the positioning with GNSS (GPS, Galileo, etc.), and the detection of conventional signalling balises (ASFA) are used.

Currently, in satellite positioning the uncertainty is less than 3 m 99% of the time with coverage, although it can reach values less than 1 m. In fact, this positioning, with the service EGNOS SoL (Safety of Life), meets aviation standards and is used in landing maneuvers [22,23].

The TBM contains the position of the virtual marks (VMs) distributed along the route. These VMs are located at points where the orography allows better satellite coverage. The density of VM in the map is a function of the proximity to the points where there are signs. If there is no signalling, the positioning error is not relevant. However, at the points where there are signals that condition the actions of the train it is important to have more precision in positioning. VMs allow the position to be corrected using satellite positioning. This procedure is carried out from 1 km before the situation of the VM until the position of the VM is exceeded. The distance between the current train position (*X_i_)* and the position of the virtual mark (*X_VM_*), called *D2VM,* is approximated by the line that joins the current train position to the mark using the Equation (1).
(1)$$D2VM=Arc\left(X_{i}-X_{VM}\right)\times R,$$
where *X_i_* are the GNSS coordinates of the train at the moment T_i_, *X_VM_* are the GNSS coordinates of the mark and *R* is the radius of the earth, Equation (2).
(2)$$R=Altitude+\sqrt{\frac{{\left(r_{1}^{2}\cdot \cos\left(Lat\right)\right)}^{2}+{\left(r_{2}^{2}\cdot \sin\left(Lat\right)\right)}^{2}}{{\left(r_{1}\cdot \cos\left(Lat\right)\right)}^{2}+{\left(r_{2}\cdot \sin\left(Lat\right)\right)}^{2}}}$$
where *r*_1_ and *r*_2_ the radius at the Equator and at the Pole at sea level, 6378.137 km and 6356.752 km respectively, and *Lat* is determined by using the Equation (3).
(3)$$Lat=\frac{Lat_{X_{i}}+Lat_{X_{VM}}}{2}$$

To avoid stopping rail traffic by the use of auscultation vehicles, the definition of the position of the virtual marks (VMs) is done on a map or with support vehicles, at points close to the track. For this reason, this position does not coincide with the exact passage of the train, so that, in the approach phase, the coordinates of the virtual marks (VMs) are transformed into those corresponding to the projections of these on the trajectory of the train.

The procedure, in that case, consists of the following steps (see Figure 6):The last five positions provided by the satellite positioning system, *X_i-4_* to *X_i_* in Figure 6, are recorded at 1-s intervals. To reduce the effect of positioning uncertainty, if the distance between consecutive positions is less than 30 m, then 30 m of distance between consecutive positions are fixed.The train’s trajectory is approximated to a straight line using least squares (blue line in Figure 6).The last known train position (*X_i_*) obtained by GNSS and the positioning mark *X_VM_* are projected onto the calculated line (red points in Figure 6), obtaining *X_iP_* and *X_VMP_*, respectively. Then, the distance between the two projections, *X_iP_* and *X_VMP_*, is calculated by obtaining D2VM (see Figure 6).

4.If *D2VM* is less than one kilometer, the longitudinal position of the train is corrected using the Equation (4):


(4)
KPT=KPVM−D2VM


5.If a *PB* is detected, the kilometer point of the train position on the track (*KPT*) is immediately updated using Equation (5):


(5)
KPT=KPB


6.If the kilometer point of the train position on the track (*KPT*) is greater than or equal to the kilometer point of the virtual balise position on the track (KPVB), the VB is generated.

Figure 7 shows a schematic of the operations performed when the train is en route to: (i) correct positioning, (ii) detect PBs to create DTBM, and (iii) generate VBs.

## 5. Results

To verify the operation of the system, several tests have been conducted. This section describes the tests performed to validate the operation of the software and hardware that constitute the ASAB-VB system proposed in this paper. Figure 8 shows a picture of the test platform used for testing the ASAB-VB system. The platform is made up of the following components: the ASFA system (onboard indications), the balise detector and odometric information receiver, and the ASAB-VB system.

Due to the difficulty of performing tests on the railway track with the complete equipment, tests have been performed with a balise emulator connected to the real equipment in Figure 8. Tests have also been performed on trains using inertial navigation and GNSS positioning, the results shown below correspond to these tests.

A GNSS receiver with EGNOS correction onboard the train was used for the tests and the differential correction was performed with a ground station. To determine the position with higher resolution at which the system generates the VBs, the GNSS position is linearly interpolated from the before and after coordinates captured at the instant of VB generation. These are the coordinates presented in Table 1 and Table 2. The latency of the system is 1 ms, which allows the generation of VBs with a position resolution of 5.5 cm at a speed of 200 km/h.

As an example, a test in which a virtual balise (VB) is generated is detailed. This test involves the VB to be generated and three VMs located at different distances from the VB. Figure 9 presents the location of the three VMs and the VB. It is observed that the greater the distance between the VB and the VM, the greater the error in the position at which the VB is generated with respect to the position stored in the TBM.

Table 1 presents the locations of the VMs and the VB corresponding to this test, corresponding to the route between Alcalá de Henares and Guadalajara shown in Figure 10. The kilometer point (KP), the distance to the VB (D2VB), and the GNSS coordinates (latitude and longitude) are provided in this table.

Several tests have been performed to generate the VB using the three VMs as reference. In these tests, the KPT is obtained with the inertial system.

In the first test, only the VM_2_ mark is used to perform the positioning. Figure 11 presents the last positions of the train before passing VM_2_. The blue asterisk represents the projection on the estimated train path (linear approximation) of the last position detected by the GNSS before the VM_2_ mark. The green asterisk represents the position of VM_2_ and the red asterisk is its projection on the estimated trajectory of the train.

The information at that point is:(6)$$D2VM_{2}=23.56\;m\;KPT=KPVM_{2}-D2VM_{2}=38.642-0.02356=38.61844\;D2VB=0.833+0.02356=0.85656\;\mathrm{km}$$

From that moment on, the train uses the inertial system to keep the KPT updated, and when it reaches point 39.475 it generates the VB. Table 2 presents the data recorded at the time of VB generation:

Only the kilometric point to virtual balise (KPVB) stored in the TBM is used to generate the VB. In this case, the coordinates (lat and lon) of the VB have been used to determine the positioning error. This error is calculated as the distance between the projections of the train position (Xi_P_) (black asterisk in Figure 12) and the VB (KPVB_P_) (red asterisk in Figure 12) on the linear trajectory of the train (Figure 12), at the time when the balise is generated.

Table 3 presents the errors obtained in the other test using VM_1_ and VM_3_ to perform the positioning.

As can be seen, the distance between the VMs and the VB is determinant for the error in the position at which the VB is generated. The shorter distance between VM and VB, the smaller the error. It should be considered that GPS coverage limitations in many points of the railway route make it impossible to place the VMs close to the VBs since the VMs must be located at points where there is good GPS coverage.

Finally, it should be noted that several tests were conducted with similar results. Tests were performed between September and November in different weather conditions on the railway line shown in Figure 10. During the tests, VBs and VMs were placed at different locations. It should be noted that the key aspect for accurate of VB generation (minimum error) is not the railway line length, but the maximum distance between the VMs and VBs. Therefore, the results obtained are extrapolated to railway lines of any length.

## 6. Conclusions

The ASAB-VB system developed and presented in this paper highlights the advantages it can bring to rail traffic by introducing very significant improvements concerning the performance of the ASFA digital system. All of this is achieved by introducing minimal modifications to the onboard system and without introducing any modification to the track infrastructures. In addition, the proposed system makes the location of the VBs more flexible to replace or complement the information of the PBs. Thus, the system allows loading the VBs coordinates that establish the speed limits (temporary speed limits, TSL) at different points along the route. This makes railway signaling more flexible, dynamic and safe. Additionally, the system may also be used to obtain information on the status of the physical balises (operational balises, absence of balises, damaged ones) with the consequent maintenance cost saving.

To do this, it is only necessary to accurately record the coordinates and kilometer point of the location of each of the balises (PBs and VBs) and some auxiliary points called virtual marks (VMs), that allow the system integration errors to be reset on the route. In this way, thanks to the VMs, the error in the position at which the VBs are to be generated is less than 1 m. This error is completely acceptable for this type of application.

It is important to note that a substantial number of references based on the use of VBs in the context of ERMS signaling systems can be found in the literature. However, the proposed ASAB-VB system is intended for conventional lines with low traffic density where ERTMS is not available. Therefore, the concept of VB is different in this application, where VBs are used to facilitate the incorporation of temporary speed limits (TSL) into the railway track. TSL is used in unexpected risk situations: falling objects on the track, unscheduled maintenance operations, landslides, unexpected accidents, etc.

The operation of the system is conditioned to the detection of a PB or GNSS coverage in the vicinity of the point where the VB is to be located. As shown in the results section, for distances of 1.5 km, the position error is less than 1 m. The accuracies shown in the experiment are calculated using differential GNSS correction with a ground station. It must be considered that if there is no such possibility during the journey, the error would be increased by the error provided by the GNSS itself with EGNOS correction, which as mentioned above, is less than 3 m 99% of the time. This error combined with that of the odometric-inertial system, is still sufficient for the location of temporary speed limitation VBs (which are the main application). If the correction is made with PBs, the results would be the same as shown. It is not easy to quantify the level of safety provided by the ASAB-VB system. For this purpose, information should be obtained over a long period of time to compare the number of accidents before and after the implementation of the ASAB-VB system. Qualitatively, the ASAB-VB system represents an unquestionable improvement in safety. Improvement in the safety is due to:In the ASB-VB system, VBs are used to facilitate the incorporation of temporary speed limits (TSL) into the railway track. TSL is used in unexpected risk situations: falling objects on the track, unscheduled maintenance operations, landslides, accidents, etc.The system makes it possible to identify broken down or damaged physical balises (PB). These broken down or damaged balises could cause accidents. In addition, damaged balises are detected when trains are in service, i.e., without the need for specific maintenance operations.

Future lines of research include (i) adding redundancy and diversity to the system to increase its reliability, and (ii) complementing the VM generation based on the information provided by GNSS with MV generation based on information obtained from natural and artificial marks (poles, traffic lights, trees, signs, etc.). This avoids dependence on GNSS coverage. Artificial vision systems could be used to detect natural and artificial marks.

## 7. Patents

The result from the work reported in this manuscript (ASAB-VB system) is patented in Spain with patent number ES 2418929 A1.

## Figures and Tables

**Figure 1 sensors-22-01943-f001:**
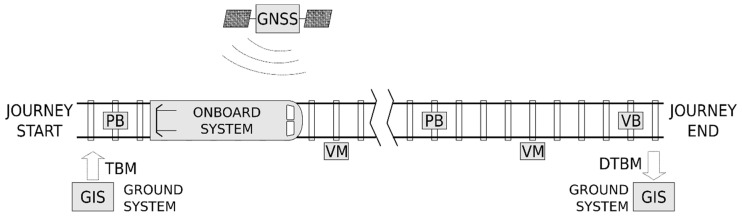
General configuration of the ASAB-VB system.

**Figure 2 sensors-22-01943-f002:**
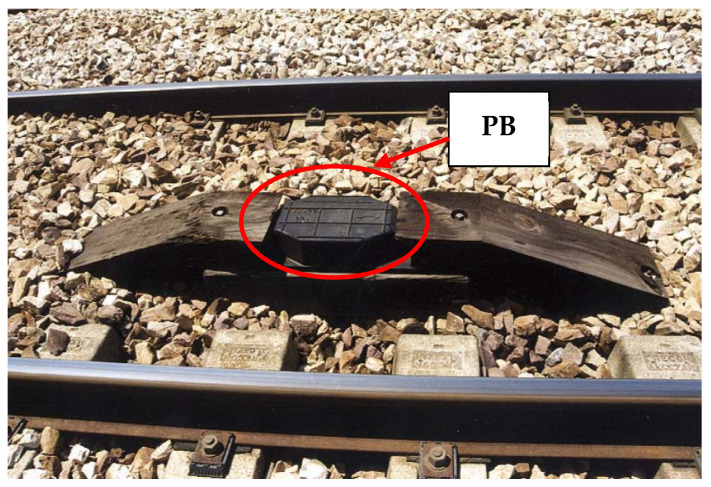
General configuration of the ASAB-VB system.

**Figure 3 sensors-22-01943-f003:**
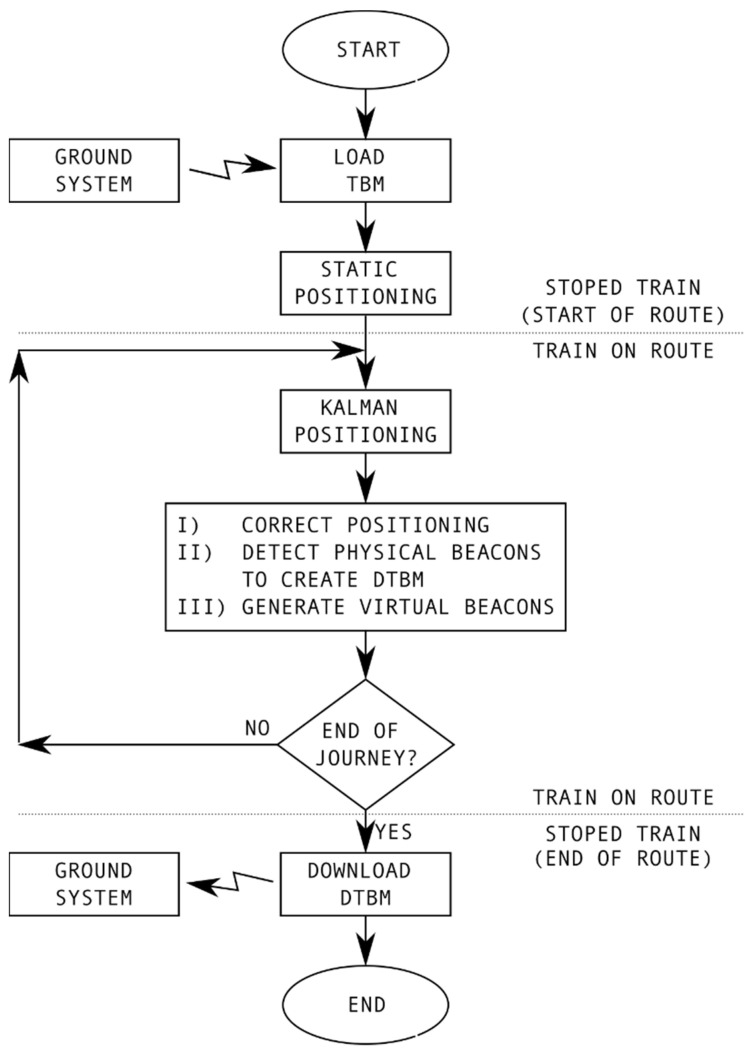
The general procedure of the proposed ASAB-VB system.

**Figure 4 sensors-22-01943-f004:**
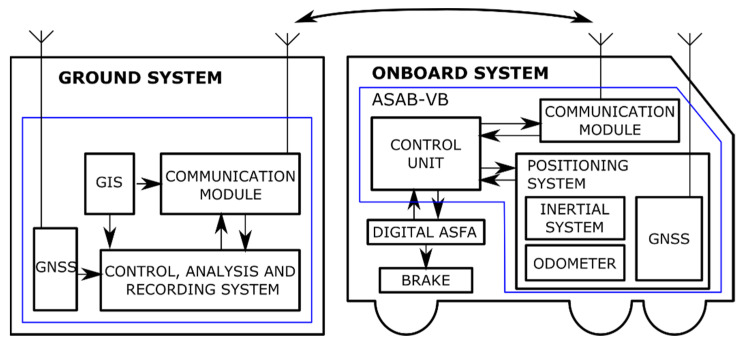
The architecture of the ASAB-VB system.

**Figure 5 sensors-22-01943-f005:**
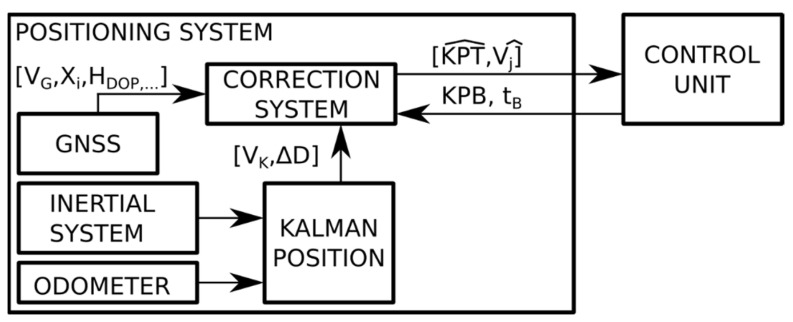
General block diagram of the positioning system onboard the train.

**Figure 6 sensors-22-01943-f006:**
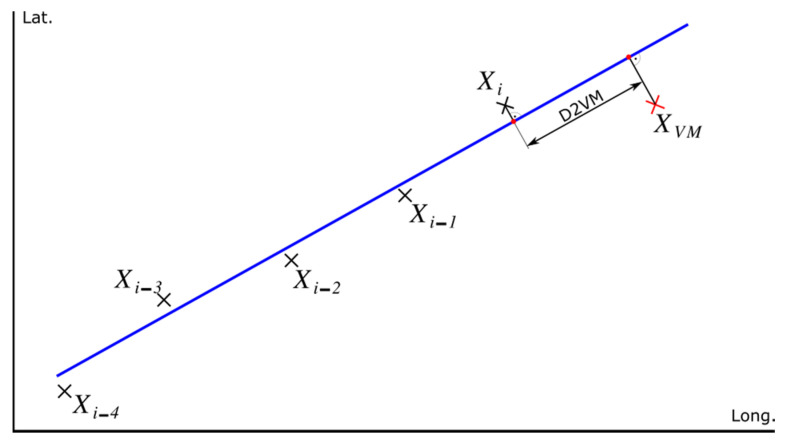
Train position correction using virtual marks (VM).

**Figure 7 sensors-22-01943-f007:**
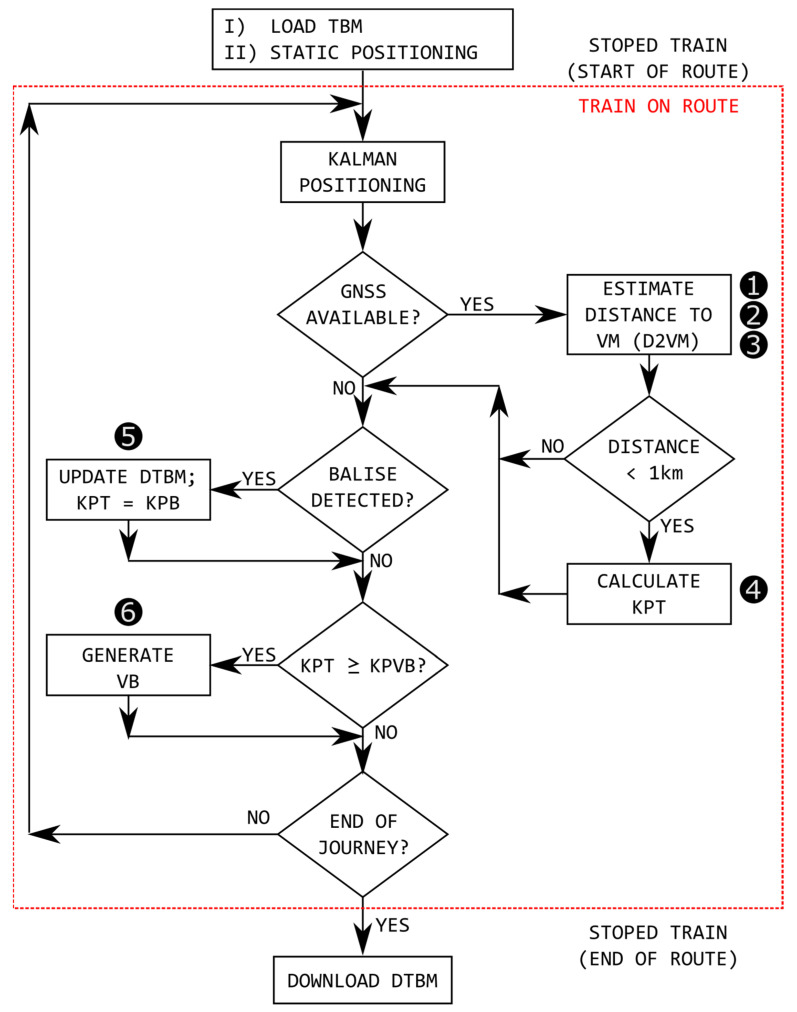
Operations performed to (i) correct positioning, (ii) detect PBs to create DTBM, and (iii) generate VBs (train on route).

**Figure 8 sensors-22-01943-f008:**
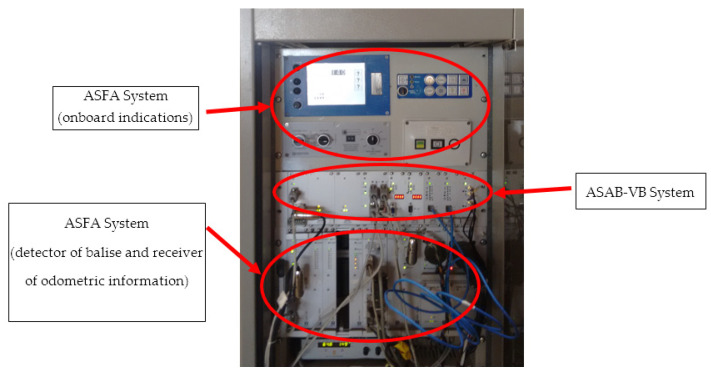
Test platform.

**Figure 9 sensors-22-01943-f009:**
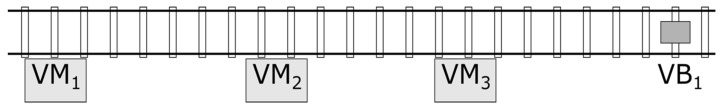
VMs and VB in the test.

**Figure 10 sensors-22-01943-f010:**
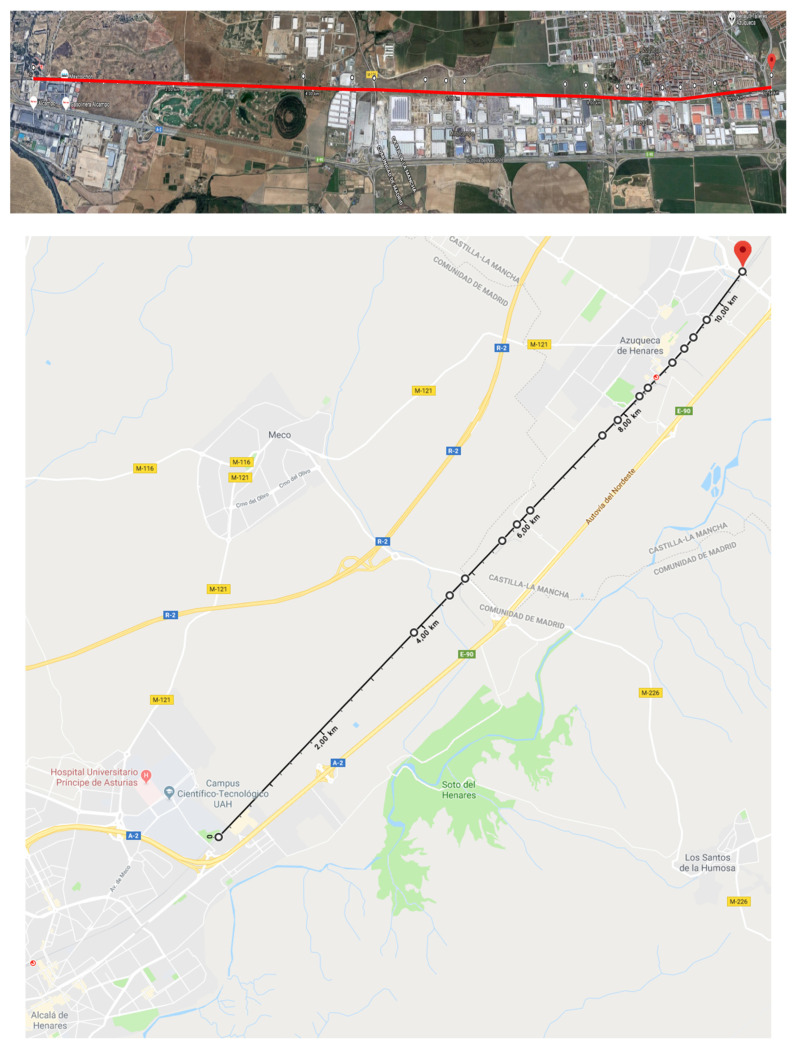
Map of the route with the location of the PBs (https://www.google.es/maps/ (accessed on 8 November 2021)).

**Figure 11 sensors-22-01943-f011:**
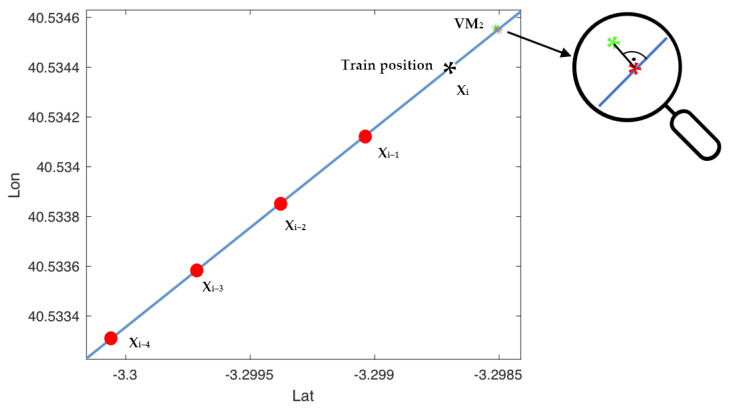
MV and train position locations in the test.

**Figure 12 sensors-22-01943-f012:**
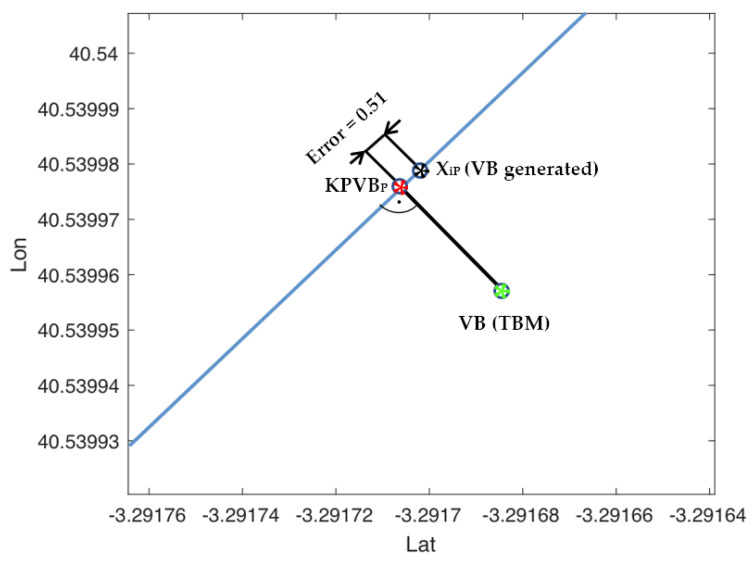
Position error in the generation of VB. VB position according to TBM (green asterisk), VB projection on the linear trajectory of the train (red asterisk), and VB generation point (black asterisk).

**Table 1 sensors-22-01943-t001:** VB and VM data.

	KP (km)	D2VB (km)	Lat (°)	Lon (°)
VB_1_	39.475	0	N-40.539961	W-3.291680
VM_1_	37.895	1.580	N-40.529683	W-3.304593
VM_2_	38.642	0.833	N-40.534558	W-3.298513
VM_3_	39.052	0.423	N-40.537163	W-3.295077

**Table 2 sensors-22-01943-t002:** VB generation data.

Lat (°)	Lon (°)	Error (m)
N-40.5399758	W-3.2917061	0.51

**Table 3 sensors-22-01943-t003:** Results summary.

	Lat (°)	Lon (°)	Error (m)
VM_1_	N-40.5399730	W-3.2917095	0.92
VM_2_	N-40.5399758	W-3.2917061	0.51
VM_3_	N-40.5399812	W-3.2916992	0.33

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
