# Peer review of "Announcement Signals and Automatic Braking Using Virtual Balises in Railway Transport Systems"

_sensors, 2022, doi:10.3390/s22051943_

Round 1

Reviewer 1 Report

This paper presents an idea of using virtual balises for ASAB. The idea is not new, the VB applications have been raised and applied in various projects, this has been mentioned by the author. The methodology of this paper is not quite suitable for publishing on a journal. Since the paper concerns the error of the VB generated data. The accuracy, latency and capture mechanism should be analyzed more in detail instead of calculations and analysis in table 2 and 3.

Author Response

Firstly, we would like to thank the reviewer for their contribution. The suggestions were very useful during the revision process and have been incorporated into the revised paper.

This paper presents an idea of using virtual balises for ASAB. The idea is not new, the VB applications have been raised and applied in various projects, this has been mentioned by the author. The methodology of this paper is not quite suitable for publishing on a journal.

As the reviewer indicates, the idea of using virtual beacons is not new, although this is true when ERTMS is the signaling system. As indicated in line 102: “The concept of virtual balise (VB) arises in the context of ERTMS to obtain an improved ERTMS”. However, there are no published solutions to improve safety on low traffic density line layouts where virtual beacons are not used. As indicated in line 108 “The ASAB-VB system is aimed at conventional lines with low traffic density where ERTMS is not available. Therefore, the concept of VB is different in this application, where VBs are used to facilitate the incorporation of Temporary Speed Limits (TSL) into the railway track”. TSLs are used in unexpected risk situations: falling objects on the track, unscheduled maintenance operations, landslides, accidents, etc. Therefore, this proposal allows the incorporation of virtual balises at points along the railway where it is necessary to temporarily carry out automatic protection actions.

Another important contribution is the incorporation of Virtual Marks (VMs) used to correct the error accumulated by the odometric system (line [120]). There are no previous works using techniques based on VMs to correct positioning errors.

In summary, numerous references based on the use of VBs in the context of ERMS signaling systems can be found in the literature. However, as indicated in line [125]: “No previous works are proposing the use of VBs and VMs for the purposes indicated in this paper”.

Since the paper concerns the error of the VB generated data. The accuracy, latency and capture mechanism should be analyzed more in detail instead of calculations and analysis in table 2 and 3.

To carry out the tests, a GNSS receiver with EGNOS correction was used on board the train and the differential correction was performed with a ground station. To determine with higher resolution the position at which the system generates the VBs, the GNSS position is linearly interpolated from the before and after coordinates captured at the instant of VB generation. These are the coordinates presented in tables 1 and 2. The latency of the system is 1ms, which allows the generation of BVs with a position resolution of 5.5 cm at a speed of 200 km/h.

In this regard and with the aim of providing more information, we have included the above explanatory texts from line 437.

Reviewer 2 Report

Page 1, Keywords: My suggestion is to include “ASAB-VB” in the Keywords since “Announcement Signals and Automatic Braking using Virtual Balises (ASAB-VB)” has already a patent number.

Section 1. Introduction, lines 32-34: My suggestion is to provide some more safety statistics as far as railway safety is concerned (especially when compared to other transport systems like road transport, air transport, maritime transport etc.).

Page 2, line 85: Please note that reference [15] appears after reference [7]. References should appear in numerical order. Therefore, please correct accordingly the respective references.

Page 4, line 151, Physical Balises (PBs): My suggestion is to include some photographs of types of PBs for the benefit of the reader.

Section 2. General overview of the proposal: As far as positioning is concerned, could you please include within the text some more details concerning the case where the train passes through a tunnel?

Page 5, line 375, (…..several tests have been conducted.): Please include within the manuscript some more details concerning the tests (e.g., time period).

Page 11, Figure 8. Situación de baliza y marcas en el ensayo descrito.: Please write the specific heading in English.

Page 11, Tabla 1. VB and VM data.: Please change “Tabla” to “Table”.

Page 11, Tabla 1. VB and VM data.: Please write n.a. (not available or not applicable) in the empty cell (column 3, row 2).

Page 12, Figure 9. Situación de la VM y las posiciones del tren en el ensayo.: Please write the specific heading in English.

Page 12, Tabla 2. VB generation data.: Please change “Tabla” to “Table”.

Page 12, equation (9): Please note that equation (9) must be equation (6).

Page 13, Tabla 3. Results summary.: Please change “Tabla” to “Table”.

Page 13, line 444, “…several tests were performed with similar results: .): Please include within the manuscript some more details concerning these tests.

Page 14, Figure 11. Map of the route with the location of the PBs.: Please include the source of the geographical background (map).

Page 14, Section 6. Conclusions: Please try to address each one of your recommendations to the respective stakeholders’ departments/organizations. Please also include the directions for future research on the topic of your work. Is it possible to provide an estimate concerning the % improvement of the overall railway safety level due to the introduction of the ASAB-VB system? Finally, please include a summary with the constraints and limitations of your work.

Page 15, 7. Patents: My suggestion is to change “7. Patents” to “Patents”.

Page 15, Reference List, [9]: Please delete [X1].

Page 16, Reference List: Please use capital letters for the first letter of the month in references [15], [17], [19], [20] and [21].

Author Response

Firstly, we would like to thank the reviewer for their contribution. The suggestions were very useful during the revision process and have been incorporated into the revised paper.

We greatly appreciate the fact that the reviewer highlighted the mistakes and we welcome your suggestions.

  • Page 1, Keywords: My suggestion is to include “ASAB-VB” in the Keywords since “Announcement Signals and Automatic Braking using Virtual Balises (ASAB-VB)” has already a patent number.

  • Section 1. Introduction, lines 32-34: My suggestion is to provide some more safety statistics as far as railway safety is concerned (especially when compared to other transport systems like road transport, air transport, maritime transport etc.).

Thanks to the incorporation of new technologies, train crashes are becoming rarer in the European Union, recently published Eurostat data provides an overview of the positive trend. In 2019, there were 516 significant railway accidents with 802 fatalities in the EU, a decrease on the 853 deaths recorded in 2018 [1].

According to “Report on Railway Safety and Interoperability in the EU-2020” [2], the fatality risk for a train passenger is one-fourth of the risk for a bus/coach passenger. The use of individual transport means, such as passenger car carries a substantially higher fatality risk:   car occupants have almost 50 times higher likelihood of dying compared to train passengers traveling over the same distance. The fatality risk for an average train passenger is now about 0.05 fatalities per billion passenger kilometers, making it comparatively the safest mode of land transport in the EU.

In this regard and with the aim of providing more information, we have included the above explanatory texts from line 37.

  • Page 2, line 85: Please note that reference [15] appears after reference [7]. References should appear in numerical order. Therefore, please correct accordingly the respective references.

  • Page 4, line 151, Physical Balises (PBs): My suggestion is to include some photographs of types of PBs for the benefit of the reader. A photograph has been added showing the appearance of a PB installed on the track.

It should be noted that the external appearance of the balises is the same regardless of their type or “balise appearance”. The train's balise detection system consists of a resonant circuit, which oscillates at a certain frequency (Permanent Frequency, PF). When the detector passes over a balise, it oscillates at the resonant frequency of the circuit set in the beacon, with which it is tuned by inductive coupling, receiving the corresponding " balise appearance ".

In this regard and with the aim of providing more information, we have included the above explanatory texts from line 169.

Section 2. General overview of the proposal: As far as positioning is concerned, could you please include within the text some more details concerning the case where the train passes through a tunnel? In the case of tunnels, in the absence of GPS coverage, the positioning system is based on the information provided by the odometric system. It should be noted that in most tunnels there are physical beacons that allow correcting the odometric error, a VM can be positioned at the closest point to the entrance of the tunnel at a point where GPS coverage is good. Correcting the odometric error when GPS coverage is expected to be lost in the vicinity of the VMs is precisely the purpose of VMs. 

In this regard and with the aim of providing more information, we have included the above explanatory texts from line 340.

  • Page 5, line 375, (…..several tests have been conducted.): Please include within the manuscript some more details concerning the tests (e.g., time period). Tests were performed between September and November in different weather conditions on the railway line shown in Figure 10. During the tests, VBs and VMs were placed at different locations. It should be noted that the key aspect for accurate of VB generation (minimum error) is not the railway line length, but the maximum distance between the VMs and VBs. Therefore, the results obtained are extrapolated to railway lines of any length.

In this regard and with the aim of providing more information, we have included the above explanatory texts from line 505.

  • Page 11, Figure 8. Situación de baliza y marcas en el ensayo descrito.: Please write the specific heading in English.

  • Page 11, Tabla 1. VB and VM data.: Please change “Tabla” to “Table”.

  • Page 11, Tabla 1. VB and VM data.: Please write n.a. (not available or not applicable) in the empty cell (column 3, row 2). Zero has been indicated. In the case of VB, the parameter D2VB=0.

  • Page 12, Figure 9. Situación de la VM y las posiciones del tren en el ensayo.: Please write the specific heading in English.

  • Page 12, Tabla 2. VB generation data.: Please change “Tabla” to “Table”.

  • Page 12, equation (9): Please note that equation (9) must be equation (6).

  • Page 13, Tabla 3. Results summary.: Please change “Tabla” to “Table”.

  • Page 13, line 444, “…several tests were performed with similar results: .): Please include within the manuscript some more details concerning these tests.

To carry out the tests, a GNSS receiver with EGNOS correction was used on board the train and the differential correction was performed with a ground station. To determine with higher resolution the position at which the system generates the VBs, the GNSS position is linearly interpolated from the before and after coordinates captured at the instant of VB generation. These are the coordinates presented in tables 1 and 2. The latency of the system is 1ms, which allows the generation of BVs with a position resolution of 5.5 cm at a speed of 200 km/h.

In this regard and with the aim of providing more information, we have included the above explanatory texts from line 437.

  • Page 14, Figure 11. Map of the route with the location of the PBs.: Please include the source of the geographical background (map).

  • Page 14, Section 6. Conclusions: Please try to address each one of your recommendations to the respective stakeholders’ departments/organizations. The organization responsible for the railway infrastructure in Spain, ADIF, was one of the partners in the project in which the ASAB_VB system was developed. Therefore, this company is aware of the results obtained.

  • Please also include the directions for future research on the topic of your work. Future lines of research include i) adding redundancy and diversity to the system to increase its reliability, ii) complementing the VM generation based on the information provided by GNSS with MV generation based on information obtained from natural and artificial marks (poles, traffic lights, trees, signs, etc.). This avoids dependence on GNSS coverage. Artificial vision systems will be used to detect natural and artificial marks.

In this regard and with the aim of providing more information, we have included the above explanatory texts in Conclusions section from line 556.

  • Is it possible to provide an estimate concerning the % improvement of the overall railway safety level due to the introduction of the ASAB-VB system?

It is not easy to quantify the level of safety provided by the ASAB-VB system. For this purpose, information should be obtained over a long period of time to compare the number of accidents before and after the implementation of the ASAB-VB system. Qualitatively, the ASAB-VB system represents an unquestionable improvement in safety.  Improvement in the safety is due to:

  1. In ASB-VB system, VBs are used to facilitate the incorporation of Temporary Speed Limits (TSL) into the railway track”. TSL is used in unexpected risk situations: falling objects on the track, unscheduled maintenance operations, landslides, accidents, etc.
  2. The system makes it possible to identify broken down or damaged physical balises (PB). These broken down or damaged balises could cause accidents. In addition, damaged balises are detected when trains are in service, i.e. without the need for specific maintenance operations.

In this regard and with the aim of providing more information, we have included the above explanatory texts in line 543.

  • Finally, please include a summary with the constraints and limitations of your work.

The operation of the system is conditioned to the detection of a PB or GNSS coverage in the vicinity of the point where the VB is to be located. As shown in the results section, for distances of 1.5 km, the position error is less than 1 m. The accuracies shown in the experiment are calculated using differential GNSS correction with a ground station. It must be considered that if during the journey there is no such possibility, the error would be increased by the error provided by the GNSS itself with EGNOS correction, which as mentioned above is less than 3 m 99% of the time. This error combined with that of the odometric-inertial system, is still sufficient for the location of temporary speed limitation VB (which are the main application). If the correction is made with PBs, the results would be the same as shown.

In this regard and with the aim of providing more information, we have included the above explanatory texts in line 534.

  • Page 15, 7. Patents: My suggestion is to change “7. Patents” to “Patents”.

  • Page 15, Reference List, [9]: Please delete [X1].

  • Page 16, Reference List: Please use capital letters for the first letter of the month in references [15], [17], [19], [20] and [21].

Reviewer 3 Report

The paper clearly gives the environment, assumptions, requirements and objectives of the problem in hand, and points out major issues or difficulties when dealing with the problem and the system design. In the paper, the performance evaluation is also presented, which makes the paper a complete work. The paper is well-organized, but it can be improved as follows:

  1. This work incorporates virtual balises (VB) in the points of track where it is necessary to temporarily carry out automatic protection actions. This work also considers the position error in the generation of VB. To further assess the system feasibility, it would be better to carry out a thorough safety study.
  2. Please check captions of Figs. 8 and 9.
  3. The authors claim that the proposed system allows the incorporation of VB that complement and/or replace the PBs. Is it feasible to consider both the virtual signaling and physical implementation simultaneously, which also complies with high safety requirements?

Author Response

Firstly, we would like to thank the reviewer for their contribution. The suggestions were very useful during the revision process and have been incorporated into the revised paper.

The paper clearly gives the environment, assumptions, requirements and objectives of the problem in hand, and points out major issues or difficulties when dealing with the problem and the system design. In the paper, the performance evaluation is also presented, which makes the paper a complete work. The paper is well-organized, but it can be improved as follows:

  1. This work incorporates virtual balises (VB) in the points of track where it is necessary to temporarily carry out automatic protection actions. This work also considers the position error in the generation of VB. To further assess the system feasibility, it would be better to carry out a thorough safety study.

It is not easy to quantify the level of safety provided by the ASAB-VB system. For this purpose, information should be obtained over a long period of time to compare the number of accidents before and after the implementation of the ASAB-VB system. Qualitatively, the ASAB-VB system represents an unquestionable improvement in safety.  Improvement in the safety is due to:

  1. In ASB-VB system, VBs are used to facilitate the incorporation of Temporary Speed Limits (TSL) into the railway track”. TSL is used in unexpected risk situations: falling objects on the track, unscheduled maintenance operations, landslides, unexpected accidents, etc.
  2. The system makes it possible to identify broken down or damaged physical balises (PB). These broken down or damaged balises could cause accidents. In addition, damaged balises are detected when trains are in service, i.e. without the need for specific maintenance operations.

In this regard and with the aim of providing more information, we have included the above explanatory texts in line 543.

  1. Please check captions of Figs. 8 and 9. Done

  1. The authors claim that the proposed system allows the incorporation of VB that complement and/or replace the PBs. Is it feasible to consider both the virtual signaling and physical implementation simultaneously, which also complies with high safety requirements?

Yes, it is possible feasible to consider both the virtual signaling and physical implementation simultaneously, resulting in a higher level of safety.

In fact, both in the patent and in the paper [line 123] it is indicated that the system may also be used to obtain information on the status of the PVs (operational balises, absence of balises, damaged ones). Detection of damaged or malfunctioning PBs prevents rail accidents.
